# Burn Guidelines—An International Comparison

Katharina I. Koyro * , Alperen S. Bingoel, Florian Bucher and Peter M. Vogt

Department of Plastic and Reconstructive Surgery, Hand Surgery, Burn Center, Medizinische Hochschule Hannover, Carl-Neuberg-Straße 1, 30625 Hannover, Germany; Bingoel.alperen@mh-hannover.de (A.S.B.); bucher.florian@mh-hannover.de (F.B.); Vogt.peter@mh-hannover.de (P.M.V.)

\* Correspondence: koyro.katharina@mh-hannover.de; Tel.: +49-(0)511-5328-894

**Abstract:** Burn injuries can be life-threatening, thus standardized procedures are essential to ensure the best medical care is provided after injury. Therefore, burn care guidelines were created throughout the world. There are many similarities within the different burn guidelines, especially in basic burn care procedures. Taking a closer look, it becomes clear that there are also a lot of disparities within the guidelines. In this review the guidelines of the German Society of Burn Treatment (DGV), British Burn Association (BBA), European Burns Association (EBA), American Burn Association (ABA), Australian and New Zealand Burn Association (ANZBA), and the International Society for Burn Injuries (ISBI) are compared. The DGV-guidelines focus on pre-hospital treatment measures, intensive care treatment and acute wound therapy, whereas the BBA puts emphasis on infrastructure and staff qualification. The EBA created guidelines for medical practitioners and non-medical staff to standardize burn care in European countries with special focus on clear treatment recommendations and best infrastructural facilities. The ABA underlines the need for best qualified medical staff and ABLS- (Advanced Burn Life Support) standards. The ANZBA focuses on best treatment options including novel wound healing biotechnologies and post-burn return-to-function rehabilitation. In contrast to all other guidelines, the ISBI does not only deal with burn care in developed countries but also in resource-limited settings. Special focus lies on the discussion of ethical issues and cost-effectiveness. In this review, advantages and disadvantages of each guideline are discussed. These findings are supposed to help improving burn care procedures worldwide.

**Keywords:** burn care; guideline; burns; comparison; burn association; international agencies

## 1. Introduction

Clinical practice guidelines are based on highly qualified literature and offer systematic recommendations designed to help medical staff in decision-making about appropriate care and management for specific clinical circumstances. Burn injuries require a long-term treatment schedule due to the complexity of the multifaceted clinical course. Patients need accurate treatment beginning with the initial injury and acute life-sustaining measures, clinical stabilization processes including best intensive care treatment and conservative and surgical wound management and reconstruction, followed by long-term rehabilitation. The treatment should take place in an appropriate setting depending on the severity of the burn injury. Severe burns should be transferred to a burn center. Additionally, infrastructural conditions and staff qualification are vital for burn care [1]. To optimize these processes, many countries have designed their own clinical practice guidelines to offer a standard for best medical care in all clinical phases. Whereas there are no great disparities in the basic steps in burn treatment worldwide, different burn associations focus on specific aspects concerning first aid procedures, transfer criteria to a burn center, treatment- and rehabilitation- recommendations.

Burn care guidelines have been written by the DGV [2–6], BBA [7–11], EBA [12], ABA [13–31], ANZBA [32], and ISBI [33,34].

In this publication, similarities and differences and the subsequent advantages and disadvantages of each guideline are discussed in detail. The aim of this publication is to improve and standardize burn care guidelines.

## 2. Materials

The DGV-guidelines are composed of five individual documents all of which are only available in German, thus they are applicable in German-speaking countries. The DGV offers separate guidelines for burn care treatment in adults and pediatric patients. Both have a clear and precise structure. The *AWMF-S2K-guideline for the treatment of burn injuries in adults* was updated in 2021, the *AWMF-S2K-guideline for the treatment of burn injuries in childhood* in 2015. The three remaining documents focus on staff, institutional, room and equipment requirements, as well as resources in burn centers. Additionally, first aid for burn injuries and recommendations for rehabilitation are outlined. All documents are freely available on the internet [2–6].

The BBA-guidelines consist of multiple documents that are free of charge and available on the British Burn Association homepage. The main document *National Burn Care review—Standards and Strategy for Burn Care* was written in 2001 [7]. Multiple additional documents can be found on the homepage (Status as of 05/2021), for instance *First aid clinical practice guidelines, Management of burns in pre-hospital trauma care, Nutritional services, Standards of physiotherapy and occupational therapy practice in the management of burn injured adults and children,* and the *National burn care referral guidance* [8–11].

The EBA-guidelines are available in English and consist of one single document and are free of charge. The latest revision was made in 2017 [12]. EBA-guidelines find their application in all European countries aside from German-speaking countries as they follow DGV-guidelines.

The ABA-practice guidelines are published in the Journal of Burn Care and Research and are written in English [13–30,35]. The guidelines are available complimentary to ABA-Members whereas non-members must purchase them online. There are multiple additional documents available on the ABA-homepage such as the *Initial first aid treatment for minor burns* recommendations (accessed on 29 May 2021). The referral criteria are listed on the ABA-homepage and can also be found in the *Guidelines for trauma centers caring for burn patients* [36]. The Advanced Burn Life Support (ABLS)-Provider manual was updated in 2018 [31]. The ABLS-Provider manual is not part of the ABA-guidelines, but as the ABLS program is described to be the *ABA's premier educational resource for emergency care of burn injuries* it was included in this review (https://ameriburn.org/education/abls-program/, accessed on 29 May 2021).

The ANZBA-guidelines are a single document updated in 2014 [32]. Allied heath practice guidelines are assisted by informative images and written in English. They must be purchased. A discount for ANZBA-members is applied. Multiple concise documents regarding burn care recommendations such as *ANZBA referral criteria, Initial management of severe burns, Pharmacy advice,* or *First aid* can be found free of charge on the ANZBA-homepage (accessed on 29 May 2021).

The ISBI-guidelines are available in English, Spanish, and Arabic. ISBI-practice guidelines consist of two parts [33,34]. Part 1 was published in *Burns* in 2016. Part 2 is an extension of part 1 addressing different aspects of burn care treatment and was published in *Burns* in 2018. Part 1 is open access, whereas part 2 must be purchased. The ISBI addresses burn care recommendations for resource-abundant and resource-limited countries.

In addition to the recommendations given in the guidelines, information on the homepage of the different burn associations is included in this analysis.

## 3. Results

In this section, the similarities and differences concerning first aid measures, transfer criteria to a burn center, inpatient treatment and rehabilitation in the recommendations of the DGV, BBA, EBA, ABA, ANZBA, and ISBI are analyzed.

### 3.1. First Aid Procedures

When burn injuries occur, first aid measures are important. All burn associations offer first aid recommendations until the appropriate treatment is initiated by medical staff. The recommendations for first aid procedures are summarized in Table 1. All burn associations recommend a similar initial assessment and the removal of clothing and jewelry [6,8,12,34] (https://anzba.org.au/care/first-aid/, http://ameriburn.org/wp-content/uploads/20 17/05/burnfirstaid.pdf, accessed on 29 May 2021). Some additionally include information about electrical injuries. Hypothermia in burn-injured patients must be avoided as hypothermia is a risk factor for a severe progression [37–40]. Cooling burn injuries to reduce pain is listed in all first aid recommendations. All burn associations do not recommend using iced water. However, the recommendations differ in the procedure of cooling. BBA-, EBA-, ANZBA-, and ISBI- guidelines recommend the cooling the burn with running water for approximately 20 min [8,12,34] (https://anzba.org.au/care/first-aid/, accessed on 29 May 2021). The DGV only recommends cooling with tap water in small burn injuries and includes exclusion criteria for cooling [6]. The ABA recommends at least 5 min of cooling for first-degree burns (http://ameriburn.org/wp-content/uploads/20 19/08/first-aid-fact-sheet-2.pdf, accessed on 29 May 2021). All burn associations suggest a simple burn dressing. DGV-, BBA-, ANZBA-, and ISBI-Association offer first aid recommendations for chemical burns [6,8,34] (https://anzba.org.au/care/first-aid/, accessed on 29 May 2021). While all burn associations except for ABA do not differentiate their first aid measures regarding the degree of the burn injury, the ABA explains how to identify the degree of burn injury and offers instructions for each degree (http://ameriburn.org/wp-content/uploads/2017/05/burnfirstaid.pdf, accessed on 29 May 2021).

### 3.2. Transfer Criteria to a Burn Center

To provide a good quality of medical care in burn-injured patients an accurate assessment and the correct treatment of the patient in the most appropriate setting is necessary. Therefore, an organized system of burn care transfer is necessary. According to the degree of severity, burn-injured patients are treated in different settings. Less severe burn injuries are treated in emergency departments, a burn facility, or a burn unit. Most burn associations do not provide detailed information about the requirements for these burn wards. Burn centers (BC) provide the highest quality expertise and most specialized resources to offer the best treatment and recovery for severely burn-injured patients [41]. However, not all patients need this level of specialized burn care. Thus, transfer criteria to a burn center are an important tool. The transfer criteria to a burn center differ in the examined guidelines.

Whereas all mentioned guidelines offer recommendations for the transfer to a burn center, only the BBA differentiates their criteria about when a transfer to a burn facility, a burn unit, or the burn center should take place [10]. According to the BBA a burn facility can be a standard plastic surgery ward, where non-complex burn injuries are treated. A burn unit is designed for patients with a moderate level of burn injury complexity. The ward is separately staffed and access to a nearby operating theater is required. In the burn center the highest level of burn injury complexity is treated. It consists of a separately staffed ward and offers a direct access to an operating theater. The burn center ward is up to an intensive care unit level of critical care [7]. Therefore, the thresholds for referral are complex, in particular because BBA distinguishes the transfer criteria to a burn facility, -unit and -center between children and adults. The BBA recommends a number of medical conditions as strict referral criteria, while others should be further discussed in order to determine if a transfer is needed [10]. In this review only strict referral criteria to a burn center were mentioned.

**Table 1.** First aid procedures.

| Guideline | DGV | BBA | EBA | ABA | ANZBA | ISBI |
|---|---|---|---|---|---|---|
| Initial assessment | Personal safety Stop the burning process Prevention of hypothermia | Personal safety Stop the burning process Prevention of hypothermia | Prevention of hypothermia | Stop the burning process | Personal safety Stop burning process, turn off electrical current Prevention of hypothermia | Personal safety Remove the subject from burning source Prevention of hypothermia |
| Clothing | Remove clothing and jewelry | Remove clothing and jewelry | Remove clothing and jewelry | Remove clothing and jewelry | Remove clothing and jewelry | Remove clothing and jewelry |
| Cooling | Cooling of small burn injuries Cooling of burn wounds is not recommended - if TBSA is >5% - in children, if torso or head are involved - in unconscious patients | Cooling the burn wound for 20 min | Cooling the burn wound for 20 min | Cooling the first-degree burn wound for >5 min | Cooling the burn wound for 20 min | Cooling the burn wound for 15–20 min |
| Dressing | Cover the burn wound in simple burn wound dressing | Cover the burn wound non-adherently Cover non-burned areas | Cover the burn wound non-adherently | First-degree burns: Cover the burn wound in a clean dressing and apply soothing lotions with aloe vera | Cover the burn wound in a clean dressing | Cover the burn wound in a clean dressing |
| Further measures | Irrigation of chemical burns | Irrigation of chemical burns after removal of chemical agents Cool tar and bitumen burns, remove tar Rewarm cold burns continually Treat electrical burns by ATLS standard, cooling, and monitoring | | First-degree burns: Drink fluids in case patient is dehydrated Over-the-counter pain reliever Second-degree: <7 cm: treat like minor burn >7 cm or involvement of feet, face, eyes, ears, groin, or major joints: see a family doctor or emergency room Third-degree: Immediately contact health care provider | Irrigation of chemical burns after removal of chemical agents Limb elevation in circumferential burns | Irrigation of chemical burns after removal of chemical agents Limb elevation during transport to limit edema Electrical injuries: Responder safety, turn off electric source, evaluation if cardiopulmonary resuscitation is necessary, cooling burn wounds Inhalation injury: Nursing of the patient in a semi-upright position with moderate elevation of the head and trunk |

In Table 2 the referral criteria to a burn center of each guideline were assorted and presented in brief. The main differences between the criteria concern the depth and percentage of TBSA of burns required for referral. Besides the BBA-recommendations, the EBA offers the most detailed transfer criteria especially for children regarding the burn depth and burned TBSA [12]. ISBI-guidelines are the least detailed. The ISBI mainly recommends to evaluate the burn by TBSA and quote ABA-referral criteria [33]. All guidelines except for BBA-guidelines outline the need for transfer to a burn center, if certain body regions such as hands, face, or genitals are burned. Some also include circumferential burns, burns at major joints, or inhalation injuries. The BBA also specifically suggests a transfer to a burn unit when special body areas are burned, but not to a burn center [10]. Special mechanisms for the burn injury such as chemical or electrical burns are also recommended to be transferred in some guidelines. Some differences concerning the transfer are due to patient-specific characteristics, such as comorbidities or severe trauma. Unlike the other guidelines, the ANZBA also recommends a transfer for pregnant women [32]. The EBA suggests referral for patients with diseases which have the clinical presentation, complications, and treatment requirements of burn injuries, such as toxic epidermal necrolysis, etc. [12].

### 3.3. Treatment Recommendations

All burn associations give similar basic treatment recommendations, such as first aid measures, analgesia, basic wound treatment, and inhalation injury procedures. In addition, every burn association puts special emphasis on certain key aspects (Table 3).

The DGV focuses on the management of life-threatening conditions. ABCDE primary survey in the trauma room and concise treatment recommendations for resuscitation and intensive care treatment, for example fluid management, are outlined. For fluid management the Parkland/Baxter formula is proposed. Wound therapy depends on burn depth. Indications for conservative and surgical therapy, such as tangential and epifascial necrectomy, are discussed [2]. Detailed treatment recommendations for children both pre-hospital treatment and conservative and surgical inpatient care are outlined separately in the pediatric guideline [3].

The BBA in particular emphasizes the differences in the structure of burn ward options. The personnel and infrastructural needs for a burn facility, a burn unit, and a burn center for the best treatment of burn-injured patients are presented in detail. Furthermore, treatment recommendations for the management of burns in pre-hospital trauma care are highlighted. However, little information in acute inpatient care treatment is given [7,9,11].

The EBA focuses on staff and infrastructural qualifications. Requirements concerning the burn center and personnel qualifications on medical and non-medical staff are comprehensively described. The management of a burn shock is described in detail. Especially fluid management with Parkland/Baxter and modified Brook formula and the use of vasopressors and inotropic agents are outlined [12]. However, other treatment recommendations, such as surgical therapy, are less precise compared to DGV-, ANZBA- or ISBI-guidelines. In particular, therapy options based on the degree of burn are not mentioned.

The ABA highlights pre-hospital and acute life-saving treatment recommendations [13,17–19,21–23,26,27,29]. Aside from a comprehensive document about burn shock resuscitation, ABLS- (Advanced Burn Life Support) treatment is recommended for emergency care in burn injuries. The separate ABLS-Provider manual focuses on the treatment procedures in the first hours after burn injuries [31]. For instance, ABCDE primary survey for initial assessment, shock and fluid resuscitation are outlined. Traditional fluid resuscitation formulas such as Parkland/Baxter or modified Brook formula are mentioned. Initial and adjusted fluid rates for children and adults depending on the mechanism of burn (thermal, chemical, or high-voltage electrical) are proposed [31]. Recommendations for different etiologies for burn wounds such as electrical-, chemical-, radiation-, cold-, and blast-injuries are also listed. The ABLS-provider manual makes recommendations concerning burn mass casualty incidents and disaster management. In addition, the ABA offers multiple separate practice guidelines for specific aspects of the treatment of burn injuries such as practice guidelines for the management of acute pain or escharotomy and decompressive therapies in burns [17,21].

**Table 2.** Transfer criteria to a burn center.

| Guideline | DGV | BBA | EBA | ABA | ANZBA | ISBI |
|---|---|---|---|---|---|---|
| Depth/TBSA | Second-degree burns >10% TBSA in children and adults<br>All third-degree burns in adults<br>Third-degree burns >5% TBSA in children<br>All fourth-degree burns in children | All degree burns >40% or >25% TBSA with inhalation injury in adults<br>All degree burns >30% TBSA in children >1 year<br>All degree burns >15% TBSA in children <1 year<br>Third-degree burns >20% TBSA in children | All degree burns:<br>>5% TBSA in children under 2 years<br>>10% TBSA in children 3–10 years<br>>15% TBSA in children 10–15 years<br>>20% TBSA in adults<br>>10% TBSA in seniors over 65 years<br>Deep partial thickness burns and full thickness burns in any age group and any extent | Partial thickness burns >10% TBSA<br>All third-degree burns | All degree burns >10% TBSA<br>All degree burns >5% TBSA in children<br>Full thickness burns >5% TBSA | Second-degree burns >10% TBSA<br>All third-degree burns |
| Specific body regions | Hands, face, genitals<br>Inhalation injury | | Hands, face, genitals, major joints<br>Inhalation injury<br>All circumferential burns | Hands, face, genitals, perineum, feet, major joints<br>Inhalation injury | Hands, face, genitals, perineum, feet, major joints<br>Inhalation injury<br>Circumferential limb or chest burns | Hands, face, genitals, perineum, major joints |
| Specific mechanism | Chemical burns<br>Electrical burns<br>Lightening burns | | Major chemical burns<br>Major electrical burns | Chemical burns<br>Electrical burns<br>Lightning burns | Chemical burns<br>Electrical burns | High-voltage electrical burns |
| Specific patients | Burn patients:<br>-With comorbidities<br>-With injuries complicating the treatment<br>-With special psychological, psychiatric, or physical needs | Burn patients:<br>-With major trauma<br>-Assessed as requiring end of life care (discuss transfer to BC vs. local palliative care)<br>Children with burn injuries:<br>-Predicted to require ventilation for more than 24 h<br>-Who are physiologically unstable | Burn patients:<br>-Requiring burn shock resuscitation<br>-Requiring special social, emotional, or long-term rehabilitation support<br>-With concomitant trauma or diseases complicating the treatment, prolong recovery or affect mortality<br>-With diseases requiring treatment in a burn center (e.g., toxic epidermal necrolysis, necrotizing fasciitis, staphylococcal scalded skin syndrome etc.), if the TBSA is >10% in children and elderly and >15% in adults | Burn patients:<br>-With diseases complicating the management, prolong recovery or affect mortality<br>-With concomitant trauma in which the burn injury poses the greatest risk of morbidity or mortality<br>-Who require special social, emotional, or rehabilitative intervention<br>Burns in children | Burn patients:<br>-With pre-existing illness<br>-With major trauma<br>-Who are pregnant<br>-Who are very young or elderly<br>-With non-accidental burns | |

**Table 3.** Initial and inpatient treatment recommendations.

| Guideline | DGV | BBA | EBA | ABA | ANZBA | ISBI |
|---|---|---|---|---|---|---|
| | Pre-hospital trauma care<br>Trauma room management<br>Analgosedation<br>Resuscitation<br>Ventilation<br>Nourishment<br>Anti-infective therapy<br>Analgesia<br>Burn wound therapy depends on degree of burn:<br>- First-degree: Conservative treatment<br>- Second-degree (a): Debridement and adequate dressing<br>- Second-degree (b): Debridement, tangential necrectomy, removal of necrotic tissue, split skin grafts<br>- Third-degree: Debridement, tangential and/or epifascial necrectomy, split skin grafts, temporary artificial skin grafts<br>- >2/3 circumferential: Escharotomy<br>Psychological care<br>Pediatric treatment recommendations in separate guideline | Pre-hospital trauma care:<br>- Airway, breathing, circulation<br>- Temperature management<br>- Burn severity<br>- Cooling<br>- Chemical burns<br>- Burn dressing<br>- Fluid resuscitation<br>- Analgesia<br>- Safeguarding<br>- Escharotomy<br>Overview of the seven phases of burn management:<br>- Rescue<br>- Resuscitate<br>- Retrieve<br>- Resurface<br>- Rehabilitate<br>- Reconstruct<br>- Review | Staffing requirements with job description for:<br>- Nurses<br>- Physiotherapists<br>- Psychologists<br>- Dieticians<br>- Social workers<br>- Occupational therapists/ergo therapists, speech therapists<br>- Pediatric care<br>- Educational therapists<br>Treatment recommendations for:<br>- Nursing (nutrition, analgesia, fluid resuscitation, wound care)<br>- Physiotherapy/occupational therapy (edema management, splinting, positioning, scars exercise, mobilization, hand rehabilitation)<br>- Pediatric care<br>Practice guidelines for burn practitioners:<br>- Initial management of burn wounds<br>- Burn wound dressings<br>- Management of burn shock | Advanced Burn Life Support (ABLS) treatment for the first 24 h post-injury:<br>- Initial assessment<br>- Airway management, inhalation injury<br>- Shock and fluid resuscitation<br>- Burn wound management<br>- Electrical and chemical injuries<br>- Pediatric burn injuries<br>- Stabilization, transfer, transport<br>- Burn disaster management<br>- Glasgow coma scale<br>- Tetanus prophylaxis<br>- Radiation-, cold-, blast injuries<br>Many specific separate ABA-practice guidelines, i.e.,<br>- Burn shock resuscitation [13]<br>- Surgical and non-surgical wound care under austere conditions [27]<br>- Surgical management of the burn wound and use of skin substitutes [26]<br>- Management of acute pain [21]<br>- Escharotomy and decompressive therapies [17] | Common areas of practice:<br>- Burn assessment<br>- Inhalation injury<br>- TBSA<br>- Zones of injury<br>- Infection control<br>- Staff support<br>- Self-care etc.<br>Dressing biotechnology in burns<br>- Topical antimicrobials<br>- Hydrocolloids<br>- Alginate<br>- Foams<br>- Hydrogels<br>- Topical negative pressure wound therapy etc.<br>Surgery in burns<br>Surgical biotechnology and acute wound reconstruction:<br>- Artificial skin dressings and xenografts<br>- Cultured epithelial autograft<br>- Dermal templates or scaffolds<br>Pain management<br>Pediatric management | Inhalation injury: diagnosis and treatment<br>Burn shock resuscitation<br>Escharotomy and fasciotomy<br>Wound care, topical agents<br>Surgical management of burn scars<br>Infection prevention and control<br>Antibiotic stewardship<br>Nutrition<br>Analgesia<br>Sedation<br>Management of comorbidities: Sepsis, pneumonia, urinary tract infections, thrombosis, psychiatric disorders etc.<br>Electrical and chemical burns |

The ANZBA offers highly structured treatment recommendations, starting with recommendations concerning general diagnostic procedures, followed by dressing biotechnology, surgery, and surgical biotechnology in burns. Therapy recommendations are based on the degree of the burn injury. Besides common wound therapy options and reconstruction of tissue defects with split skin grafts and different flap techniques, treatment recommendations for the use of artificial skin dressings and xenografts, cultured epithelial autografts and dermal templates are explained in detail. Images facilitate the understanding [32]. Thus, ANZBA-guidelines deal more with inpatient than pre-hospital treatment and burn shock resuscitation. For instance, fluid resuscitation formulas are not described.

The ISBI presents treatment recommendations for pre-hospital treatment including resuscitation, fluid management, tetanus immunization, and inpatient care. A special focus lies on the applicability under all medical conditions. The recommendations address developed countries as well as countries with resource-limited settings. Evaluations of benefits and harms, values and preferences and costs are listed. Treatment recommendations for comorbidities such as pneumonia or urinary tract infections are included in the guidelines [33,34].

### 3.4. Rehabilitation Recommendations

All burn associations itemize rehabilitation recommendations. However, the recommendations differ greatly (Table 4). The DGV offers rehabilitation guidelines with a clear structure involving criteria for the indication of rehabilitation, different types of rehabilitation programs and concise treatment recommendations [4]. The BBA and EBA focus more on psychological aspects in rehabilitation than on treatment recommendations for long-term burn injury associated medical conditions [7,12]. The EBA highlights return-to-work and school procedures. The ABA offers multiple documents on the homepage containing rehabilitation recommendations. Some are written as concise practice guidelines for a defined medical condition [14,16,20,21,26,27,30,42,43]. The ABA also offers the *Burn rehabilitation and research: Proceedings of a consensus summit* manuscript, summarizing clinical burn care rehabilitation [24]. The ANZBA describes detailed rehabilitation procedures. Clear and precise treatment recommendations are given and return-to-function recommendations are outlined [32]. The ISBI only mentions rehabilitation recommendations briefly and is the least precise [33,34].

**Table 4.** Rehabilitation recommendations.

| Guideline | DGV | BBA | EBA | ABA | ANZBA | ISBI |
|---|---|---|---|---|---|---|
| | Indications for rehabilitation after burn injuries<br>Personnel conditions to attend a rehabilitation program<br>Different types of rehabilitation programs<br>Duration of rehabilitation<br>Rehabilitation centers<br>Focus of rehabilitation:<br>- Treatment of scars<br>- Nursing in rehabilitation<br>- Movement therapy<br>- Treatment of contractures<br>- Psychological care<br>- Analgesia<br>- Comorbidities<br>- Amputations<br>- Social rehabilitation<br>- Cooperation burn center and rehabilitation center<br>- Therapy of long-term effects<br>Detailed pediatric rehabilitation guidelines | Psycho-social rehabilitation<br>Physical therapy<br>Psychological support<br>Scar modulation<br>Psychiatric support<br>Rehabilitation provision<br>Quantification of rehabilitation need<br>Continuing care model for burn injury<br>Clinical networks for burn injury | Preparations for discharge from a burn center including a discharge checklist<br>Psycho-social guidelines for:<br>- Anxiety<br>- Depression<br>- Delirium<br>- Quality of life<br>- Return-to-work<br>- Working with parents/siblings<br>- Back to school | Multiple practice guidelines, i.e.:<br>Cardiovascular fitness [14]<br>Silicone [42]<br>Early ambulation after lower extremity grafts [16]<br>Burn rehabilitation therapist competency tool [43]<br>Burn rehabilitation and research: Proceedings of a consensus summit [24]:<br>- Administrative issues and initiatives<br>- Research and Education<br>- Documentation<br>- Hand burns<br>- Exercise in burn patient management<br>- Burn patient perioperative rehabilitation management<br>- Splinting and casting<br>- Edema<br>- Positioning<br>- Burn Scar<br>- Pain/Pruritus<br>- Physical agents to manage burn scar<br>- Outcome of burn survivors<br>- Head and neck burns<br>- Critical care aspects | Measuring post-burn recovery<br>Edema management<br>Exercise and mobility<br>Return-to-function<br>Splinting and positioning<br>Scar management<br>Orofacial contracture management<br>Psychosocial management | Positioning the burn patient in positions to prevent contractures<br>Splinting of the burn patient<br>Maintain or promote movement and physical function<br>Pruritus management |

## 4. Discussion and Conclusions

Intensive heat contact to the skin can cause burn injuries. In addition to thermal burns due to flames, scalding or hot vapors, electric-and chemical burns can lead to severe burn injuries needing a specialized and multidisciplinary therapy [44]. Furthermore, some diseases such as toxic epidermal necrolysis are associated to burn injuries and are usually treated in burn centers [45]. The therapy of burn injuries can be divided in different stages of burn care. In the beginning acute life-saving procedures and if applicable intensive care treatment procedures are paramount. After stabilization, wound management is essential. Secondary consequences of burn injuries can be intense, thus post-burn rehabilitation procedures are important to address the long-term sequelae. Inappropriate treatment at any stage may exert adverse effects on the subsequent course of the injury and may affect restoration of body function and reintegration in everyday life.

### 4.1. Recent Updates

There are many different burn associations worldwide trying to set standards in all stages of burn care. In 2016 a comparison of the DGV-, ABA- and EBA-guidelines was published [46]. Since then, EBA-, DGV-, and ABA-guidelines and the ABLS-Provider manual were updated in 2017, 2018, 2020 and 2021 as medical technology continues to advance. For instance, the transfer criteria to a burn center in the latest DGV-guidelines (2021) are stricter than in the guidelines published in 2010 [2]. During the latest revision of the EBA-guidelines (2017) first aid measures were included for the first time [12]. The ABA recently updated their management of pain recommendations [21]. This shows that research in burn injuries is constantly ongoing and new findings are added to the guidelines. However, the burn associations have different priorities that make it difficult to decide how burn injuries should be treated and decide what is really important in burn care.

### 4.2. Advantages and Disadvantages of the Guidelines

In this review, the similarities and differences of the latest recommendations of six well-known burn associations (DGV, BBA, EBA, ABA, ANZBA, and ISBI) were analyzed. The goal of this review was to outline the advantages and disadvantages of each guideline to improve each of them. First aid measures, transfer criteria to a burn center, inpatient treatment and rehabilitation were compared and discussed. In Table 5 a number of advantages and disadvantages of each guideline were summarized.

The DGV offers detailed and well-structured first aid, referral, treatment, and rehabilitation recommendations and focuses on burn center requirements. It is worthy of note to mention the separate comprehensive guidelines for pediatric patients. As a disadvantage, no English version is currently available. Therefore, DGV-guidelines are only relevant in German-speaking countries and not internationally. Aside from this, the DGV mainly focuses on recommendations for medical staff. As burn injuries require multidisciplinary care, recommendations for non-medical staff would be desirable.

The BBA identifies the differences between burn facilities, -units and, -centers. The referral criteria are the most complex as the BBA provides separate thresholds separately for children and adults to a burn facility, -unit, and -center. Additional to the strict referral criteria, separate criteria as well as when a transfer should be discussed, are also listed. These detailed criteria for transfer might be an advantage, however, in emergency situations simplified criteria could save time. Acute inpatient care could be described in more detail.

EBA-guidelines are written as a single document carefully designed for quick reference. The definition, conditions and function of a burn center are described in detail. However, the referral criteria to a burn center are complex, as they do not only depend on burned TBSA but also age. More simplified criteria would be desirable. Another advantage of EBA-guidelines is the focus on multidisciplinary burn care treatment as they offer recommendations for medical staff as well as non-medical staff. For instance, the guidelines address recommendations for nurses, physiotherapists, psychologists, dieticians, social workers, occupational and speech therapists, educational therapists, as well as others.

**Table 5.** Advantages and disadvantages of the guidelines.

| Guideline | DGV | BBA | EBA | ABA | ANZBA | ISBI |
|---|---|---|---|---|---|---|
| Structure/Layout | Five separate documents, easy to find on the homepage | Multiple documents | Single document | Multiple documents | Single document | Two documents |
| Costs | Free | Free | Free | Free for ABA-Members, Non-Members must purchase the guidelines | Discount for ANZBA-Members, Non-Members must purchase the guidelines | Published in *Burns,* Part 1 open access, Part 2 purchase necessary |
| Language | German | English | English | English | English | English, Spanish, Arabic |
| Advantages | - Concise and **well-structured** first-aid, referral, treatment, and rehabilitation recommendations<br>- Recommendations for **demands of the burn center**<br>- Separate **guideline for pediatrics** | - Most detailed **definition of burn facilities, burn units and burn centers**<br>- **Transfer criteria** differentiate between transfer to a burn facility, -unit, or -center | - **Concise** and well-structured recommendations for **demands of the burn center**<br>- Focus on **infrastructural and staff requirements** | - **ABLS- Provider manual** for structured first aid and initial treatment procedures<br>- Very **detailed in specific issues**, such as first aid procedures, management of acute pain etc. | - Single **well-structured document** with the focus on inpatient treatment recommendations<br>- Detailed chapter for **special biotechnology in wound dressings and surgery**<br>- **Images** for better understanding | - **Cost-effectiveness** listed<br>- **Ethical issues** are debated<br>- Addresses **resource-limited and resource-abundant settings** |
| Disadvantages | - **Non-medical staff requirements and tasks not defined in detail**<br>- **Useless in non-German speaking countries** | - **Little information about acute inpatient treatment recommendations**<br>- **Confusing because of multiple documents** | -**Complicated transfer criteria** | - **Confusing because of multiple documents without a clear structure**<br>- **Difficult accessibility** | - Acute therapy/pre-hospital management briefly mentioned | - **Briefly mentioned rehabilitation recommendations** |

As the first hours after burn injury are assumed to have a great impact on long-term outcome, the ABA recommends the participation in the Advanced Burn Life Support (ABLS)- courses. The ABLS-Provider manual is free of charge and summarizes detailed and structured first aid- and initial assessment procedures [31]. Furthermore, recommendations concerning burn mass casualty incidents and disaster management are included. However, on the ABA-homepage many different documents can be found. Some are not available for non-ABA-Members, which makes it difficult to receive all information. It would be desirable to summarize information in a single comprehensive document.

The ANZBA-guidelines are structured within a single document. The use of images allows for a better understanding of the content. ANZBA-guidelines are supposed to support medical and non-medical staff in direct patient care. The main attention lies on practical advice rather than framework conditions in burn care. The ANZBA describes the most detailed wound treatment in burn injuries. In particular, the latest dressing biotechnologies and surgical biotechnology procedures are outlined. However, this information is only helpful for burn specialists in a burn center, where these treatment measures are available. Some infrastructural advice and more information about emergency treatment recommendations could improve the ANZBA-guidelines.

The ISBI differentiates in medical and infrastructural conditions. All other guidelines mainly focus on burn care treatment in developed countries. The ISBI debates ethical issues, outlines cost-effectiveness, and evaluates implementation of burn care recommendations under resource-limited conditions. In this context, ISBI is the most applicable in developing and emerging countries. However, rising healthcare costs are also a problem in industrialized countries. Therefore, it would be advisable that all burn associations include cost-effectiveness in their recommendations. The global benefit of the ISBI-guidelines is underlined by its availability in three languages (English, Spanish, Arabic). First aid and treatment options are described in detail. More detailed rehabilitation recommendations would be beneficial as rehabilitation in burn injuries is regarded as extremely important for reintegration into everyday life.

### 4.3. Possible Reasons for the Differences in the Guidelines

There are many similarities within the different guidelines, especially concerning basic primary care procedures. However, there are also disparities. A number of these differences can be attributed to the fact that health care systems differ substantially worldwide. Others are due to a different focus on specific procedures in burn care. The differences concerning specific procedures are partly a result of different economic settings, which are addressed by the guidelines. Whereas DGV-, BBV-, EBA-, ABA-, and ANZBA mainly address high-income countries, the ISBI addresses all economic variances. Aside from this, the associations appear to target different groups, for instance the DGV, EBA, and ANZBA target burn professionals. BBA and ABA offer many documents, while some target burn professionals, others are more useful for medical staff that handle burns less frequently. The ISBI offers recommendations for both. Furthermore, most burn associations offer extra documents with recommendations (i.e., for first aid) on their homepages for patients and medical staff that handle burn injuries less frequently.

### 4.4. Conclusions

As listed in Section 4.2 this review highlights the advantages and disadvantages of the examined guidelines with the aim to improve each of them. All guidelines would benefit from being written as a single comprehensive manuscript with subsections for non-burn-specialized medical staff and burn experts in burn centers.

**Author Contributions:** K.I.K. analyzed the data. A.S.B., F.B. and P.M.V. contributed conceptional input. The manuscript was written by P.M.V. and K.I.K. All authors have read and agreed to the published version of the manuscript.

**Funding:** This research received no external funding.

**Institutional Review Board Statement:** Not applicable.

**Informed Consent Statement:** Not applicable.

**Conflicts of Interest:** The authors declare no conflict of interest.

**Websites of Burn Associations:** Some data were obtained from the homepage of each burn association (accessed on 29 May 2021):

–    DGV: https://verbrennungsmedizin.de (accessed on 29 May 2021)
–    BBA: https://www.britishburnassociation.org (accessed on 29 May 2021)
–    EBA: https://www.euroburn.org (accessed on 29 May 2021)
–    ABA: https://ameriburn.org (accessed on 29 May 2021)
–    ANZBA: https://anzba.org.au (accessed on 29 May 2021)
–    ISBI: https://www.worldburn.org (accessed on 29 May 2021)

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
