# Peer review of "Burn Guidelines—An International Comparison"

_2673-1991, doi:10.3390/ebj2030010_

Round 1

Reviewer 1 Report

This review aims to present the summary of burn guidelines used around the globe.  Overall, the review lacks impact and the information presented could be gained from reading the tables of the paper and then the guidelines themselves, there is nothing novel in the review of even why a review like this should be undertaken again.  This paper would benefit from review by a native English speaker.

Specific comments:

  1. It is American Burn Association, not Burns.
  2. It is transfer criteria, not transferal.
  3. Line 64 – what is “textbook proportions.” A lot of pages? Please reword to better describe.
  4. Line 67 – do the authors mean the actual physical facility of a burn center when they say “structural facilities?” Please reword to better describe.
  5. How is the EBA used since it is European but there is a separate guideline for European countries that speak German? This should be described. Do German speaking countries have to follow the DGV or if they speak more than German, do they follow the EBA?
  6. Line 73 – what is a high medical standard? Do all of the guidelines operate assuming a high medical standard??
  7. “Accurate wound care” on line 76 is not a proper term, please revise.
  8. Is there a reason this paper is written like a scientific paper with experiments with methods/results? The methods don’t read like methods, they read like an extension/beginning of the results.
  9. It is personnel not personal (multiple times throughout the paper, starting on line 88).
  10. The ABA guidelines are the only guidelines that you don’t share when they were last updated. You only share the update of the ABLS manual. Furthermore, the ABLS manual is part of a course, it is not part of the ABA guidelines.
  11. Jewelry is the proper spelling.
  12. Please spell out minutes in the test of the results, do not abbreviate.
  13. All tables need to be formatted such that direct comparison can be made. I suggest landscape format, smaller font. None of the tables should have periods at the end of any of the lines (style point).
  14. What is “professional management” in line 151 mean? Please revise.
  15. The difference between a burn center and burn unit needs to be explained in the paper likely around line 163.
  16. It is inhalation injury, not inhalation trauma.
  17. “Causes” of burns not a typical term, recommend etiologies or mechanisms.
  18. Skin diseases, such as TEN, have no place in this manuscript and should be removed.
  19. Degree of burn depth (line 203) is redundant.
  20. What is meant by “Preclinical?” Pre-hospital treatment? Preclinical to many people mean animal studies, please revise.
  21. “Especially highlights” (line 217 for example) is redundant.
  22. Diagnostic not diagnosis in line 230.

**23. The determination of advantages and disadvantages of each guideline are subjective, not consistent and have no real basis.  There needs to be a criteria or admission of subjective nature of the entire discussion, the latter really taking away from the point of the paper.  The authors want for example, more rehabilitation guidelines – why?  Why is it advantageous to have in depth information on novel burn wound care technologies only available at a specialized burn center and also advantageous to have a plethora of information for non-burn providers? There needs to be a clear definition of the purpose of the guidelines in the authors mind to facilitate complete revision of the discussion.**

Reviewer 2 Report

A well written review of burn guidelines from various organizations.

Language wise good, only a few minor mishaps. Maybe not that high of a scientific impact but most certainly important information and interesting comparisons. The paper is definitely worth publication.

Abstract and throughout; at least according to ameriburn.org the ABA is spelled without an “s”, ie American Burn Association, not burnS assoc.

Probably language-wise most correct, however, personally I’m somewhat annoyed by reading the word guideline, in different constellations, 119 times in the same paper (+25 in the refs). I would prefer that instead of adding “guidelines” to each organization every time maybe rephrase to eg (p1 lines 25-33): “The DGV-guidelines focus on preclinical measures, intensive care treatment and acute wound therapy, whereas BBA put emphasis on infrastructure and staff qualification. EBA highlights the importance of……and created guidelines…. ABA underline the need… ANZBA focuses on best treatment… In contrast to all other organizations ISBI do not only deal…” In this way we have already reduced the amount with 6 “guidelines”. Several more can be dropped by rephrasing.

Please add the refs to the individual papers when they’re discussed on page 2 & 3 and elsewhere. Helps when I want to look up a certain mentioned paper.

Some overlap between “Introduction” and “M&M” regarding the organizations different papers, eg “Introduction” DGV – “…guidelines are applicable….consists of five separate… The two main…”, “M&M” DGV – “The DGV-guidelines are composed of five individual….The three remaining….” Could probably be shortened and without repeating same info under 2 headings.

Is it necessary to again write out the full names of the organizations’? (“Results”) It’s the third time in same paper.

Page 6, lines 173-174 – Something’s up with this sentence, incoherent.

Page 7, Table 2, EBA, last paragraph – misspelled “scalded” (not scaled).

Page 9, Table 3, DGV, 2nd line – misspelled analgosedation.

Page 9, Table 3, EBA, 2nd paragraph – oedema is spelled edema in the other three occasions. Consistency.

Page 10, Table 3, ISBI, 6th line – should “topic agents” actually be “topical agents”?

“Discussion & Conclusion”

Even though sort of the same information is available in Table 5 I think the paper would gain by changing it to something more like Table IX in Ref #46. I do appreciate that it can be risky out of a PC context to put judgements, but it would certainly add some spice and hopefully spur the organizations to improve their guidelines.

It would also be interesting if you could add some information on how (if?) the guidelines from ref #46 have changed/improved between that paper and this one.

I also wonder whether you could add/discuss somewhat around for whom the guidelines are written. I believe there could be somewhat different groups that are targeted (ie are they written for ‘burn professionals’ or for people that handle burns less frequently) and that this may have an impact on the information in the different guidelines.

Page 14, Table 5, ANZBA, lines 9 & 12 – these do not start with capital letters as are the other lines.

“References”

The URL for ISBI needs to start with httpS otherways one ends up in a dead end.

Reviewer 3 Report

Thank you for allowing me to review the manuscript entitled: Burn guidelines – an international comparison. This article aims to improve and standardize burn care guidelines as the similarities and differences and the subsequent advantages and disadvantages of each guideline are discussed in detail including the first aid measures, transferal criteria to a burn center, inpatient treatment and rehabilitation.

It was very interesting to see how the authors mapped out the discrepancies, difference and similarities of the guidelines reviewed. Also, the language used for example the DGV is using degrees to categories the burns as opposed to the depth of injury that is more frequently used in the peer reviewed literature. In general, the manuscript is extremely word with many conjunctive adverbs such as ‘therefore’ ‘moreover’ ‘thus’ ‘furthermore’.

Page 2 line 55 basal steps should be amended to basic steps.

Table 1 on page 4 line 130 requires a capital T

Table 2 on page 6 line 168 requires a capital T

Table 3 on page 8 line 199 requires a capital T

Table 4 on page 10 line 250 requires a capital T

Table 5 on page 12 line 291 requires a capital T

Line 200 page 8: ABCDE primary survey as opposed to ABCDE- scheme

Line 221 page 8 it states ABCDE-procedure I believe you referring to primary survey.

Table 3 title is recommendations; however, it should represent the initial/acute care recommendations and needs amending. Surprisingly, fluid resuscitation formulas recommendations e.g., Parkland formula are not mentioned in the manuscript apart from the table that simply states fluid resuscitation.

The Table 1 – 5 could be presented in a format that is better organized. Currently there is a large amount of text.

The tense in the discussion is present tense when discussing the aim of the current review. However, this should be in past tense. ‘In this review the similarities and differences of the latest recommendations of six well-known burn associations (DGV, BBA, EBA, ABA, ANZBA and ISBI) are analyzed. The goal of this review is to outline advantages and disadvantages of each guideline.

The statement ‘The ANZBA-guidelines are perfectly structured in a single document’ seems to be a subjective statement given the term ‘perfectly’. It would be better to be objective and remove the term ‘perfectly’.

The use of subheadings in the discussion would structure and frame the discussion and make it easier for the reader to digest the information presented. There are no limitations presented and the conclusion is just a summary of the review not a conclusion plus the final sentence of the conclusion is misplaced.

Intext references uses parentheses not brackets. 

Reviewer 4 Report

Remarks

Abstract

 ‘The EBA-guidelines highlight the importance of the interdisciplinary work and created guidelines for the different healthcare professionals involved.’

  • I would suggest rephrasing this sentence.

Keywords

-I would suggest condensing  keywords and including less words, e.g,   burn care, guidelines, comparison.

Introduction section

  • I suggest omitting the first paragraph and mentioning only the last sentence.

‘The ABA-guidelines are well-known and served as a template for many other guidelines  internationally.’

  • I suggest omitting this sentence, instead, brief mentioning the specific features of ABA guidelines would be helpful.
  • The aim of the study should be clearly mentioned in the last paragraph. I suggest omitting the sentence ‘The aim of this publication is to improve and standardize burn care guidelines.’

Material and Methods section

  • Material and methods section should be better organized. Material and Methods section provides almost the same information with the second paragraph of the of the Introduction section. I recommend clarifying all basic specific characteristics of the guidelines used for comparison. Data from the results section (first paragraph) could be written up in the Methods section.

Results section

Table 1

‘Identifying the agent in chemical injuries, follow the specific protocol for the chemical agent, removal all contaminated clothing, irrigation up to 45 min’

  • sounds awkward, the sentence needs to be rephrased, I would recommend all suggestions be written in the same manner.

‘Inhalation injuries: position the patient between lying and sitting’

  • More precise reference to the guidelines’ recommendations would be advisable; the ISBI guidelines recommend ‘semi-upright position with moderate elevation of the head and trunk’.

- I would recommend including recently  updated ABA recommendations for the management of pain in the analysis and reference list: Journal of Burn Care & Research, Volume 41, Issue 6, November/December 2020, pp.1129–1151 instead of Ref 18 (Faucher L, Furukawa K. Practice Guidelines for the Management of Pain: J Burn Care Res. 2006 Sep;27(5):659–68.)

Section 3.2. Transferal criteria to a burn center, first paragraph

- Information provided in the first paragraph is redundant, I would recommend omitting this paragraph.

Last sentence, ‘…EBA-guidelines also suggest referral for patients with diseases associated to burns such as toxic epidermal necrolysis etc.’

- I would suggest replacing the phrase ‘diseases associated to burns’ with ‘diseases which have the clinical presentation, complications, treatment requirements and outcome of burns’ or ‘diseases mimicking a burn’.

Table 2: Transferal criteria to a burn center

  • I would suggest replacing ‘special’ with ‘specific’ in the first column
  • Last column, phrase ‘ …> 20% TBSA in adults of age’  needs editing

Table 4, last column, ISBI guidelines

It would be helpful to clarify the term ‘positioning of the burn patients.’  I would recommend adding the phrase  ‘Mobility, exercise and physical function’

Discussion section

  • First paragraph is redundant, I would suggest omitting or condensing this paragraph.

Recheck linguistic errors throughout the manuscript.

Round 2

Reviewer 1 Report

Improvements to the paper are substantial and it now reads very well and is a good review of current available guidelines.

Reviewer 3 Report

Thank you for allowing me to once again review the manuscript entitled: Burn guidelines – an international comparison. Thank you for reviewing and addressing the comments made on the previous version of the manuscript.

Keywords: Keywords need to reflex indexed terms and the subject area therefore keywords need to be amended accordingly i.e. Burns, guideline, international agencies, comparison, Burn Association.

In Table 2 on the last column, it states in blue:

‘With diseases mimicking burn injuries (toxic epidermal necrolysis, necrotizing fasciitis, staphylococcal

scalded skin syndrome etc.)’

I suggest not to say mimicking burn injuries instead state ‘diseases requiring treatment in a burns center e.g., (toxic epidermal necrolysis, necrotizing fasciitis, staphylococcal

scalded skin syndrome’.

Table 5: It states ‘pictures for better understanding’ however, the term ‘images’ may be better suited here.

Under the references there are several websites however these are not references. All references need to be formatted accordingly as per the author guidelines.

Reviewer 4 Report

As my main concerns have been addressed, I am content with the current version of the manuscript, and thus recommend publication.
